# Development and Validation of Blood-Based Predictive Biomarkers for Response to PD-1/PD-L1 Checkpoint Inhibitors: Evidence of a Universal Systemic Core of 3D Immunogenetic Profiling across Multiple Oncological Indications

**DOI:** 10.3390/cancers15102696

**Published:** 2023-05-10

**Authors:** Ewan Hunter, Matthew Salter, Ryan Powell, Ann Dring, Tarun Naithani, Maria Eleni Chatziioannou, Abel Gebregzabhar, Mutaz Issa, Jayne Green, Serene Ng, Chun Ren Lim, Cheah Soon Keat, Ang Tick Suan, Rakesh Raman, Ho Kean Fatt, Fabian Lee Wei Luen, Heba Alshaker, Dmitri Pchejetski, Dave Blum, Thomas Guiel, Robert Heaton, Jedd Levine, Alexandre Akoulitchev

**Affiliations:** 1Oxford BioDynamics Plc., Oxford OX4 2WB, UK; 2Oxford BioDynamics (M) Sdn Bhd, Penang 10470, Malaysia; 3Mount Miriam Cancer Hospital (MMCH), Penang 11200, Malaysia; cheahsoonkeat@gmail.com (C.S.K.);; 4Island Hospital, Penang 10450, Malaysia; 5School of Medicine, University of East Anglia, Norwich NR4 7TJ, UK; 6Oxford BioDynamics Inc., Gaithersburg, MD 20878, USA

**Keywords:** immuno-oncology, immune checkpoint inhibitors, response to treatment, epigenetics, blood test

## Abstract

**Simple Summary:**

Immune checkpoint inhibitors (ICIs) offer high efficacy of cancer treatment, but their use is limited to a subset of patients who respond well to the resetting of the immune system. To attempt to identify patients and predict response to ICI, tumour-based biomarkers such as PD-L1 expression or mutational tumour burden have been widely used but proved to be of insufficient accuracy. Here, we have deployed epigenetic profiling that detects specific chromosome conformations in the blood of the patients. It has been successfully used for predictive and prognostic applications. In this study, we developed and validated blood biomarkers for a checkpoint inhibitor response test that offers a significant increase in the accuracy of predicting positive response to ICI across multiple oncological indications. This new test is accurate, rapid, and minimally invasive. It could assist in treatment decisions, help to improve patient selection, and more efficiently manage costs.

**Abstract:**

Background: Unprecedented advantages in cancer treatment with immune checkpoint inhibitors (ICIs) remain limited to only a subset of patients. Systemic analyses of the regulatory 3D genome architecture linked to individual epigenetic and immunogenetic controls associated with tumour immune evasion mechanisms and immune checkpoint pathways reveal a highly prevalent molecular profile predictive of response to PD-1/PD-L1 ICIs. A clinical blood test based on a set of eight (8) 3D genomic biomarkers has been developed and validated on the basis of an observational trial to predict response to ICI therapy. Methods: The predictive eight biomarker set is derived from prospective observational clinical trials, representing 280 treatments with Pembrolizumab, Atezolizumab, Durvalumab, Nivolumab, and Avelumab in a broad range of indications: melanoma, lung, hepatocellular, renal, breast, bladder, colon, head and neck, bone, brain, lymphoma, prostate, vulvar, and cervical cancers. Results: The 3D genomic eight biomarker panel for response to immune checkpoint therapy achieved a high accuracy of 85%, sensitivity of 93%, and specificity of 82%. Conclusions: This study demonstrates that a 3D genomic approach can be used to develop a predictive clinical assay for response to PD-1/PD-L1 checkpoint inhibition in cancer patients.

## 1. Background

Insights into tumour immunology related to mechanisms of tumour immunosurveillance and anti-tumour immune responses have led to unprecedented advances in cancer treatment. This breakthrough in cancer treatment takes advantage of key mechanisms that mitigate tumour immune evasion by targeting immune checkpoints to enhance anti-tumour immunity and harnesses the overall patient immune system. These agents have shown considerable clinical benefit in certain patient populations [1,2,3].

Under physiological conditions, immune checkpoint molecules regulate the immune system by dampening the immune response following successful mitigation of an infection and preventing the onset auto-immune conditions. One of the first identified inhibitory checkpoints—CD152, also known as T lymphocyte-associated protein (CTLA-4)—has been shown to prevent expansion of CD4+ helper T cells, boost regulatory T cells, and promote a pro-tumour immuno-suppressive phenotype [4,5]. Strategies to antagonise CTLA-4 as a means of increasing anti-tumour immunity eventually lead to US Food and Drug Administration (FDA) approval of ipilimumab for treatment of metastatic melanoma [6]. With limited other therapeutic options to improve the survival of advanced melanoma patients, ipilimumab demonstrated a 2-year survival rate of 23.5%. However, consistent with auto-immunity observed in pre-clinical models targeting CTLA-4 [7,8], treatment with ipilimumab was associated with immune-related adverse effects in 60% of patients [9].

Today, the two most successfully exploited immune checkpoints are CD279, called programmed cell death protein 1 (PD-1), expressed on tumour infiltrating lymphocytes, B cells, NK cells, and myeloid cells; and its ligand, CD274, called programmed death-ligand 1 (PD-L1), expressed on tumour cells. The PD-L1/PD-1 interaction is a major mechanism leading to tumour immune evasion. Agents that interfere with this interaction have demonstrated potent and durable anti-tumour activities, with less severe immune-related toxicity compared to CTLA-4 blockade [3]. Accordingly, in earlier clinical trials the anti-PD-1 antibodies Nivolumab and Pembrolizumab had already been proven effective in treatments of melanoma, non-small cell lung cancer (NSCLC), and colorectal cancer [10,11,12,13,14], while anti-PD-L1 antibodies, such as Atezolizumab, Avelumab, Durvalumab—have been proven effective in treatments of NSCLC, urothelial carcinoma, and triple negative breast cancer [15,16,17,18]. Additional cancer types are currently under active clinical investigation.

In addition to the expanding array of therapeutic agents targeting PD-(L)1 (dostarlimab, tyvyt, libtayo, tislelizumab, camrelizumab, and sasanlimab), a series of novel immune checkpoint molecules are undergoing evaluation (LAG-3/CD223, TIM-3, TIGIT, VISTA, B7-H3/CD276, BTLA/CD272) [3].

Limitations of currently approved immune checkpoint inhibitors (ICIs) include variable responses among cancer types, primary resistance in the majority of patients with objective responses observed in the minority, acquired resistance in most cancer types, and significant risk of immune-related adverse side effects. The objective response rate (ORR) for anti-CTLA-4 ipilimumab (Yervoy) in melanoma was 10.9%, with a high-grade treatment-related adverse event rate of 15% [9]; for anti-PD-1 Pembrolizumab (Keytruda) in advanced melanoma—33% ORR, with 14% high-grade treatment-related adverse events [19]; for anti-PD-L1 Avelumab (Bavencio) in urothelial carcinoma—17% ORR, with 8% high-grade treatment adverse events [20].

In practical terms, oncologists prescribing an ICI must weigh the risk of immune-related adverse effects against the ORR and benefits of ICI treatment, with very limited data to guide the decision. This has led to multiple efforts to develop predictive biomarkers to identify patients who will benefit from treatment. The predictive value of tumour intrinsic factors such as tumour mutational burden (TMB), microsatellite instability (MSI), and DNA mismatch repair deficiencies (dMMR) has been supported by several studies [21,22,23]. In 2017, the FDA approved Pembrolizumab for treatment of advanced paediatric and adult solid tumours with high MSI and dMMR, which have not responded to prior treatments and have no other alternative treatment options [3]. The association of genetic biomarkers such as TMB with response to ICI treatment is not observed in all patients [21], and has been reported to have limited predictive value in a particular study context [24].

Also, the advanced technology and standardisation required for such biopsy-based tests impose practical limitations on applicability of these biomarkers in practice-based clinical settings.

Assessment of tumour infiltrating lymphocytes has been evaluated for predictive value with mixed success [25]. With that said, there is much work still required, one thought being that the standardisation of histologic evaluation may improve reliability [26].

Earlier trials of anti-PD-1/PD-L1 inhibitors reported association between tumour PD-L1 expression and response to treatment in melanoma and NSCLC [27,28]. In contrast, other reports showed that durable responses to ICI could be obtained in the absence of tumour PD-L1 expression [29]. The variability in the definitions of PD-L1 positivity and in methods for its evaluation account for significant inconsistencies in predictive results, with calls for further standardisation [3].

The 3D configuration of the genome plays a crucial role in coordinated gene regulation and homeostasis of cellular phenotype [30,31,32]. Three-dimensional genome architecture has been shown to act as the regulatory interface and integration point for genetic risks, epigenetic cues and modifications, metabolic signalling, and transcriptional events all integrated into the manifestation of specific cellular phenotypes and, ultimately, clinical outcomes [30,33,34]. EpiSwitch^®^ is a biomarker platform and methodology for patient stratification developed on the basis of the original chromosome conformation capture (3C) approach as a novel biomarker modality [35,36]. The EpiSwitch^®^ platform has reduced to practice all stages of the discovery, development, validation, and monitoring of blood-based biomarkers, based on 3D genome architecture. To date, 3D genomic EpiSwitch^®^ biomarkers, also known as chromosome conformation signatures; have been used in blood test format to stratify melanoma patients; prognostically stratify patients with fast versus slow progressing motor neurone disease; stratify patients with symptomatic and pre-symptomatic neurodegenerative disease; diagnose patients with thyroid cancer and various stages of prostate cancer; prognostically stratify patients for outcome in diffuse large B cell lymphoma; predictively stratify patients with NSCLC for response to the anti-PD-L1 ICI Avelumab; prognostically stratify high-risk individuals with an immune-health profile susceptible to systemic hyperreaction and severe COVID disease complications upon infection with SARS-CoV-2; and significantly increase PSA positive predictive value in the context of prostate cancer treatment in the population at risk [37,38,39,40,41,42,43,44,45,46,47,48,49,50,51].

Based on predictive and prognostic methodologies developed with systemic 3D genomic biomarkers, we looked at the EpiSwitch^®^ platform 3D genomic profiles in patients treated with ICIs to see if any could be used to predict responsiveness to ICI treatment. We have focused on the PD-(L)1 pathway target, as it is the most advanced in terms of clinically developed inhibitors. We have based our analysis on a prospective observational study with the use of several of the approved ICIs for a variety of oncological indications.

We have used the EpiSwitch^®^ Explorer array platform for whole genome profiling of patients prior to ICI treatment, with subsequent classification of the clinical outcome of response based on standard ORR criteria/RECIST 1.1, as a standard in clinical practice and trials settings [52,53]. After analysing 1.1 million data points with annotations across the whole genome for each screened patient, we identified significant and reproducible differences in marker profiles of responders and non-responders, as potential marker leads. The top leads representing alternative 3D genomic conformations were then translated into a qPCR format, evaluated, and reduced to a molecular classifier. The classifier was then validated on samples from the observational trial and from independent validation cohorts. Here, we report on the development of the 3D genomic biomarker panel with clinical utility in predicting response to ICIs targeting PD-(L)1 across a variety of oncological indications. These biomarkers reflect prevalent regulatory settings at the level of 3D genomics in the dynamic equilibrium with the patient immune system. They are systemically present at baseline prior to treatment and have predictive value for response/non-response outcomes to ICI treatments, either with a PD-1 or PD-L1 monoclonal antibody antagonist.

## 2. Materials and Methods

### 2.1. Patient Characteristics

Whole blood samples, 232 in total, were obtained from consenting patients enrolled in the observational trial “Identifying and Developing Chromosomal Conformation Signatures in Patients Undergoing Cancer Immunotherapy” at Mount Miriam Cancer Hospital (MMCH) in Penang, Malaysia. Additionally, 48 retrospective baseline IO treatment samples were procured commercially (Appendix A). Thirty-two (32) patients were used in the EpiSwitch^®^ screening and discovery stage, 77 in the training model cohort, and there were three independent validation cohorts of 24, 128, and 51 patients. The subject pool represented a multinational set of ICI treated cases, with over 40 distinct oncological diagnoses, from the United States, Europe, and Asia. Patient indications, treatments, clinical outcomes, calls by EpiSwitch^®^ classifier, and sample use at different stages of the biomarker development are listed in (Appendix A). Disease response or progression to the therapy was assessed by the investigators according to RECIST 1.1 guidelines [52].

### 2.2. Preparation of 3D Genomic Templates

EpiSwitch^®^ 3D libraries, chromosome conformation analytes converted to sequence-based tags, were prepared from frozen whole blood samples. Using EpiSwitch^®^ protocols following the manufacturer’s instructions for EpiSwitch^®^ Explorer Array kits (Oxford BioDynamics Plc, Oxford, UK), samples were processed on the Freedom EVO 200 robotic platform (Tecan Group Ltd., Männedorf, Switzerland). Briefly, 50 µL of whole blood was diluted and fixed with a formaldehyde containing EpiSwitch buffer. Density cushion centrifugation was used to purify intact nuclei. Following a short detergent-based step to permeabilise the nuclei, restriction enzyme digestion and proximity ligation were used to generate the 3D libraries. Samples were centrifuged to pellet the intact nuclei before purification with an adapted protocol from the QIAmp DNA FFPE Tissue kit (Qiagen, Hilden, Germany) Eluting in 1x TE buffer pH7.5. The 3D libraries were quantified using the Quant-iT™ Picogreen dsDNA Assay kit (Invitrogen, Waltham, MA, USA) and normalised to 5 ng/mL prior to interrogation by PCR.

### 2.3. Array Design

Custom microarrays were designed using the EpiSwitch^®^ pattern recognition algorithm, which operates on Bayesian modelling and provides a probabilistic score that a region is involved in long-range chromatin interactions. The algorithm was used to annotate the GRCh38 human genome assembly across ~1.1 million sites with the potential to form long-range chromosome conformations [29,30,31,32,33,34,35,36]. The most probable interactions were identified and filtered on a probabilistic score and proximity to the protein, long non-coding RNA, or microRNA coding sequences. Predicted interactions were limited to EpiSwitch^®^ sites greater than 10 kb and less than 300 kb apart. Repeat masking and sequence analysis was used to ensure unique marker sequences for each interaction. The EpiSwitch^®^ Explorer array (Agilent Technologies, Product Code X-HS-AC-02), containing 60-mer oligonucleotide probes, was designed to interrogate potential 3D genomic interactions. In total, 964,631 experimental probes and 2500 control probes were added to a 1 × 1 M CGH microarray slide design. The experimental probes were placed on the design in singlicate with the controls in groups of 250. The control probes consisted of six different EpiSwitch^®^ interactions that are generated during the extraction processes and used for monitoring library quality. A further four external inline control probe designs were added to detect non-human (*Arabidopsis thaliana*) spikes in DNA added during the sample labelling protocol to provide a standard curve and control for labelling. The external spike DNA consists of 400 bp ssDNA fragments from genomic regions of *A. thaliana*. Array-based comparisons were performed as described previously, with the modification of only one sample being hybridised to each array slide in the Cy3 channel [45].

### 2.4. Translation of Array-Based 3D Genomic Markers to PCR Readouts

The top array-derived markers identified in our previous study were interrogated using OBD’s proprietary primer design software package to identify genomic positions suitable for a hydrolysis probe-based real-time PCR (RT-PCR) assay [46]. Briefly, the top array-derived markers associated with predictive potential to differentiate between response and non-response to ICI outcomes were filtered on the logistic regression Glmnet coefficient. PCR primer probes were ordered from Eurofins (Luxembourg) genomics as salt-free primers. The probes were designed with a 5′ FAM fluorophore, 3′ IABkFQ quencher and an additional internal ZEN quencher and ordered from iDT (Integrated DNA Technologies, Coralville, IA, USA) [54]. Each assay was optimised using a temperature gradient PCR with an annealing temperature range from 58 to 68 °C. Individual PCR assays were tested across the temperature gradient alongside negative controls including soluble and unstructured commercial TaqMan human genomic DNA controls (Life Technologies, Carslbad, CA, USA) and used a TE buffer-only negative control. Assay performance was assessed based on Cq values and reliability of detection and efficiency based on the slope of the individual amplification curves. Assays that passed the quality criteria and presented with reliable detection differences between the pooled samples associated with responders and non-responders to ICI treatment outcomes were used to screen individual patient samples.

### 2.5. EpiSwitch^®^ PCR

Each patient sample was interrogated using RT-PCR in triplicate. Each reaction consisted of 50 ng of EpiSwitch^®^ library template, 250 mM of each of the primers, 200 mM of the hydrolysis probe, and a final 1X Kapa Probe Force Universal (Roche) concentration in a final 25 mL volume. The PCR cycling and data collection were performed using a CFX96 Touch Real-Time PCR detection system (Bio-Rad, Hercule, CA, USA). The annealing temperature of each assay was changed to the optimum temperature identified in the temperature gradients performed during translation for each assay. Otherwise, the same cycling conditions were used: 98 °C for 3 min followed by 45 cycles of 95 °C for 10 s and 20 s at the identified optimum annealing temperature. The individual well Cq values were exported from the CFX manager software after baseline and threshold value checks. A total of 20 3D genomic markers that passed the translation phase were screened on 32 responder and non-responder samples as a marker reduction step based on statistical criteria to identify the top 8 discriminating markers. These markers were evaluated on 78 individual samples from the training cohort as part of the classifier model design. They were then used to screen the independent validation cohorts of 24 and 128 samples.

### 2.6. Genomic Mapping

The 24 3D genomic markers from the statistically filtered list with the greatest and lowest abundance scores were selected for genome mapping. Mapping was carried out using Bedtools closest function for the 3 closest protein coding loci—upstream, downstream, and within the long-range chromosome interaction (Gencode v33). All markers were visualised using the EpiSwitch^®^ Analytical Portal.

### 2.7. Statistical Analysis

The 20 markers screened on 32 individual patient samples in Screens 1 and 2 were subject to permutated logistic modelling with bootstrapping for 500 data splits and non-parametric Rank Product analysis (EpiSwitch^®^ RankProd R library). Two machine learning procedures (eXtreme Gradient Boosting: XGBoost and CatBoost) were used to further reduce the feature pool and identify the most predictive/prognostic 3D genomic markers. The resulting markers were then used to build the final classifying models using CatBoost on a 78 sample cohort. Catboost is a member of the Gradient Boosted Decision Trees machine learning ensemble techniques [55]. All analysis was performed using R statistical language with the Caret, XGBoost, SHAPforxgboost, and CatBoost libraries.

### 2.8. Biological Network/Pathway Analysis

Protein interaction networks and pathway enrichment were generated using the Search Tool for the Retrieval of Interacting proteins (STRING) and Reactome Pathway Browser databases [56,57,58].

### 2.9. Causal Graph Analysis

The bnlearn (version 4.7.1) package in R (version 4.0.3) was used to generate Bayesian causal networks [59]. A score-based algorithm—the hill-climbing greedy search algorithm with bootstrapping (500 with 5 restarts)—was used as the basis for the Bayesian structure learning algorithm to generate weighting for the relationships found between the markers within the training set.

## 3. Results

### 3.1. Whole Genome Array Profiling for Discovery of Predictive 3D Genomic Marker Leads in Baseline Immuno-Oncology (IO) Patients at Baseline

The EpiSwitch^®^ array platform was used for whole genome screening and the discovery of 3D genomic biomarker leads. It has been utilised to date on over 120 IO patients, generating over 104 million individual chromosome conformation data points. We based our initial selection of marker leads on the screening results from a whole genome EpiSwitch array for 32 patients from the observational trial at Mount Miriam Cancer Hospital (MMCH). These patients were treated with either Pembrolizumab, Atezolizumab, or Durvalumab, and were diagnosed with one of the following indications: lung cancer, kidney cancer, nasopharyngeal cancer, sagittal sinus carcinoma, neuroendocrine tumour, or vulvar carcinoma. Among the responders, those patients were confirmed in the durable nature of their response and absence of acquired resistance.

From over 30 million data points, following the logistic regression Glmnet coefficient selection for baseline responders and non-responders, we identified the top 72 marker leads associated with predictive value for response and non-response to ICI (Table 1).

It is important to point out that the most significant marker in this selection was associated with CD274 and PDCD1LG2 loci, at the junction of the genes encoding for PD-L1 and PD-L2 checkpoint inhibitors. Functionally, this suggests a regulatory event associated with the predictive profile for response to ICI and leading to specific conditional differences among patients, as captured systemically through regulatory 3D genomic profiles. This is consistent with earlier observations that PD-L1 expression levels, as reflected in HIT testing, could share predictive values under specific conditions [27,28].

### 3.2. Identification of the Top Predictive 3D Genomic Markers for ICI Treatment Outcomes

Following the established methodology for EpiSwitch^®^ marker reduction [42,46], we employed a stepwise approach to translate array-based marker leads from the 32 array screened patients into qPCR format in order to identify a minimal set of biomarkers for predictive stratification of ICI treatment response outcome (Figure 1).

From over 30 million data points on array profiling, the top 72 markers were used for translation into qPCR format. The design and sequencing restrictions on the primers and fluorescent probes corresponding to the array probe sites of genome long-ranged junction points have reduced 72 markers to 24 qPCR designs (denoted in Table 1). At the experimental stage of temperature gradient optimisation, 20 qPCR marker designs have passed quality control (Table 2).

All 20 markers then underwent feature reduction in several steps (Materials and Methods). Firstly, they were evaluated in qPCR format on pooled samples from 32 patients representing either responders or non-responders. These sample cohorts represented patients treated with Pembrolizumab, Atezolizumab, and Durvalumab. Based on the results, the top 13 markers were then evaluated on patient samples individually, reducing the selection to the top eight markers (Table 2).

The eight selected markers were further evaluated on the training cohort of 77 patients (Appendix A). Blood samples taken from patients prior to initiation of ICI therapy were used together with clinical assessment for response status according to RECIST 1.1 criteria. Baseline clinical characteristics were similar between responders and non-responders (Appendix A). The training cohort of 77 patients represented ICI treatments: Pembrolizumab, Atezolizumab, and Durvalumab, with cancers of the lung, pancreas, bladder, kidney, head and neck (larynx, nasopharynx, salivary gland), liver, breast, colon, meninges, and vulva.

We used these eight markers to generate the 3D genomic classifier with predictive ability for ICI response, which we then applied to the independent test cohorts.

### 3.3. Testing of the Predictive 3D Genomic Biomarker Panel for Response to ICI Treatments on Independent Patient Cohorts

To access the predictive power of the classifier model, the eight-marker 3D genomic panel was validated on an independent baseline test cohort #1 (Appendix A). No samples from that cohort were used to rebuild or refine the model. The EpiSwitch platform readouts for the eight-marker classifier model were uploaded to the EpiSwitch Analytical Portal for analysis. Clinical outcomes for the test cohort #1 included a balanced representation of 12 responders and 8 non-responders and 3 stable diseases. It is important to mention that, from the start of the model classification design, all the stable disease cases were considered to be non-responders. EpiSwitch predictive calls based on the eight-marker model demonstrated a high performance of 83% balanced accuracy and 83% positive predictive value in the test cohort #1 (Figure 2A). Across all 101 patients used in this study in both training and testing cohorts, the test demonstrated positive predictive value of 96% and balanced accuracy of 96% (Figure 2B). The patients represented ICI treatments: Pembrolizumab, Atezolizumab, and Durvalumab, and indications including cancer of the cervix, kidney, liver, lung, neuroendocrine, meninges, and vulva.

We have further extended our validation exercise, obtaining additional 128 samples as test cohort #2 (Appendix A). This cohort was based largely on retrospective samples and represented an unbalanced group of non-responders with only nine responders among them. The EpiSwitch predictive calls based on eight-marker model demonstrated 76% accuracy, 78% sensitivity, and 76% specificity on 128 patients in the test cohort #2 (Figure 3A). Across all 229 patients used, with 170 non-responders and 59 responders in training and both testing cohorts, the test demonstrated an accuracy of 85%, sensitivity of 93%, and specificity of 82%, with a positive predictive value of 64%, and negative predictive value of 97% (Figure 3B).

The validation was lastly extended with the further collection of 51 samples exclusively from the observational trial (cohort #3, Appendix A). Unlike the retrospective collection, all the observational samples were evaluated by EpiSwitch predictive biomarkers against RECIST 1.1 response assessment preformed for the same cycle of treatment as the sample collection. The EpiSwitch predictive calls based on the eight-marker model demonstrated 87% sensitivity, 82% balanced accuracy, and 77% positive predictive value of the test cohort #3 (Figure 4A). Across all 280 samples used in this study, the test demonstrated a sensitivity of 91%, positive predictive value of 67%, and balanced accuracy of 84% (Figure 4B).

Altogether, the combined cohort of patients used in this study represented treatments with anti-PD-1 or anti-PD-L1 ICI therapies, including: Pembrolizumab, Atezolizumab, Durvalumab, and Nivolumab in a variety of indications including cancer of: pancreas, soft tissue (alveolar soft part sarcoma), bile duct (cholangiocarcinoma), bladder, cervix, vulva, kidney, head and neck (larynx, parotid gland (mucoepidermoid carcinoma), nasopharynx, oral cavity and maxilla) colon, liver, breast, lung (adenocarcinoma, small cell carcinoma, squamous cell carcinoma), lymphoepithelial carcinoma, prostate, stomach, high-grade neuroendocrine tumour, melanoma, meninges, and brain.

## 4. Discussion

Cancer treatment has been revolutionised by the development of therapies that target the immune checkpoint response [3]. However, only a minority of patients ultimately benefit from ICI therapies today. It is well recognised that there is a high prevalence of immune-related adverse events accompanying ICI treatments. Both clinical decisions and evaluation of the risk–benefit ratio would greatly benefit from the development of molecular biomarkers to predict the clinical response to therapy.

Here, we used a 3D genomics biomarker approach, which has demonstrated successful development of valuable prognostic and predictive biomarkers in oncology and autoimmune conditions [38,42,50]. We have identified eight systemic 3D genomic biomarkers that, when assessed as a molecular classifier in blood samples from a diverse group of baseline patients treated with anti-PD-(L)1 ICIs, gave an early readout of likely response or non-response to ICI therapies prior to treatment initiation.

Additionally, we have followed a subset of patients longitudinally and demonstrated that the same test re-affirms a patient’s likely response or non-response to therapy even while they are in the middle of their therapy course. Importantly, as a reflection of the network regulation, the identified 3D genomic markers are associated with genes related to the regulation of the immune system (Figure 5). This is consistent with the concept that ICI therapies exert their therapeutic effects on tumour cells by enhancing the cell-mediated immune response [3].

The captured 3D genomic differences observed between responder and non-responder baseline patients also identified 3D biomarkers corresponding to genetic loci and pathways (Figure 6) including NF-kB and TGF-b, whose biological functions are related to checkpoint response [60,61,62].

The top 3D genomic markers identified by the EpiSwitch Explorer Array profile as associated at baseline with response/non-response to ICI treatment were also analysed using the Search Tool for Retrieval of Interacting Genes (STRING) database. The view from the established protein–protein networks confirmed that the coding regions associated with the top 24 EpiSwitch marker leads are highly connected (Figure 7). In fact, no additional nodes were added to generate the network based on these 24 EpiSwitch marker leads. This is consistent with the regulatory network formed by the 3D genome architecture, as an integrator of molecular multi-omics mechanisms [30] and being concordant with controls of the protein expression and cellular phenotype.

Among other interesting observations is the high prevalence of the validated 3D marker OBD189_q057_q059 responsible for conditional regulatory long-range interactions in the region spanning CD274 (PD-L1) and PDCD1LG2 (PD-L2) (Figure 5). In fact, this systemic predictive marker has been consistently observed across multiple cohorts of IO patients, indicating that the complex network regulation manifestly shares predictive value both at the systemic 3D genomic level and at the level of PD-L1 gene expression in established IHC testing [3]. The EpiSwitch Portal view of this marker (Figure 8) identifies variable allelic frequency SNPs and the affected transcriptional factor (TF) binding motifs around the marker anchor sites. Three TFs were identified at the proximal anchor site of the marker: NFIC, ZEB1, MZF1. Analysis of the GeneHancer repository (GeneCards) shows evidence of enhancer controls (Gene association score) over CD274 (PD-L1) for the distal anchor site of the same marker. Altogether, this strongly suggests that the regulatory effect of the EpiSwitch marker over the PD-L1 locus is executed through coordinated activities of the CD274 enhancer and ZEB1 TF activities.

Finally, we deployed causal graphs, also known as Bayesian causal networks, to evaluate the relationship between the markers and other variables within the training set of the data. Such analysis is based on graphical models where nodes represent random variables and arrows represent probabilistic dependencies between them [64]. The graphical structure of a Bayesian network is a directed acyclic graph, of nodes (or vertexes) and arcs (or edges). The graph defines a factorisation of the global probability distribution, into a set of local probability distributions, one for each variable.

Analysis of the training data set showed the causal relationships, based on probabilistic relationships, between all the final EpiSwitch markers selected into the signature for the classifier model, starting from the clinical outcome of RECIST 1.1 assessment for IO treatment response (Figure 9A, RECIST 1.1 outcome). Translating the causal graph from the identities of the 3D Genomic EpiSwitch markers to the identities of the genes (Figure 9B) affected by those markers at the level of 3D genomic organisation reveals a close causal relationship between the RECIST 1.1 clinical outcome and changes in the regulatory architecture of the PD-L1 (CD274) locus, presumably contributing as a result to complex changes of its expression pattern as well. As mentioned earlier, our data suggested that the systemic 3D genomic regulatory profiles at the PD-L1 (CD274) locus shared their predictive values with threshold changes in PD-L1 gene expression from the established IHC PD-L1 testing [3].

Altogether, our data show that the predictive model developed in this study has high biological relevance and is anchored on the baseline presence of a prevalent systemic 3D genomic profile, itself functionally linked to immuno-genetic settings conducive to the clinical outcome of response to PD-(L)1 blockade. The systemic nature of the observed marker profile is not surprising since the 3D genomic approach described here was done on whole blood samples, with dominant lymphocytic representation. The fact that some T-cell-related loci were found to bear conditional regulatory differences in 3D genome architecture, in association with clinical phenotypes of responder/non-responder profiles, is a fascinating regulatory phenomenon. It is consistent with multiple observations of the role 3D genome architecture plays in regulation of oncological phenotype and clinical outcomes [30].

A simple blood-based assay that provides a readout of likely response to PD-(L)1 ICI therapy could be a valuable asset for oncologists considering ICI therapy, since only a minority of patients experience durable tumour responses. For example, ORR for metastatic NSCLC is 24% and 16%, for treatment-naïve and previously treated patients, respectively [65]. A positive PD-L1 expression result from an immunohistochemistry test, which increases ORR to 39.7% (PD-L1 50% or greater) [65]. Many of the remaining patients derive no clinical benefit, suffering significant drug-related adverse events (~14%) or even death from pulmonary toxicity (>1%) [66]. Moreover, approximately 15% of non-responders may also be at risk of hyper-progressive disease, accelerated by ICI therapy and shortening their lives by months [67]. Today, despite a low response rate, many patients are prescribed ICI therapies at a substantial financial cost to both payers and patients [68]. Altogether, these aspects highlight the importance of a response predictive assay to improve patient selection for optimised treatment, better overall treatment-decision planning, potential utilisation of alternative effective treatments, avoiding futile care and unnecessarily toxicity, and efficiently managing costs and resources [68].

This study has been exclusively focused on systemic readouts based on liquid biopsy, rather than exclusively on the solid tumour itself. Therefore, relative to predictive features for the response to immune checkpoint inhibitors, these treatments act by systemically releasing the brakes of the immune system and at the same time resetting it, all within the context of the dynamic immune exchanges between the tumour and the host. The validity of the liquid biopsy approach with 3D genomic biomarkers was initially demonstrated in the development and validation of systemic EpiSwitch biomarkers for response to Avelumab in a NSCLC cohort of 99 patients from the Javelin Solid Tumor trial NCT01772004, in collaboration with the Pfizer-EMD Serono consortium and the Mayo Clinic [50]. In that study, the systemic readout with the EpiSwitch 3D genomic profile far exceeded the predictive powers of the IHC PD-L1 expression in the tumour biopsy. Recently, the systemic EpiSwitch approach has also been validated on a cohort of 384 metastatic urothelial cancer patients tin a 2-arm study, from the Pfizer JAVELIN Bladder 100 trial NCT02603432 [69]. In that study, an EpiSwitch systemic biomarker associated with regulation of the POU2F2 locus provided highly significant binary baseline calls for benefit from treatment with Avelumab in combination with best supportive care (BSC) against the control arm of BSC alone, demonstrating a Hazard Ratio of 0.44. The systemic biomarker approach therefore significantly improved the TMB predictive classification.

Interestingly, from a broader perspective, EpiSwitch prognostic systemic biomarkers in DLBCL patients also turned out to be more informative than the punch biopsy based gene profiles classification models looking at the cell or origin distinctions (Fluidigm), particularly for Type III patient sub-type [42].

Even with such results, it might still be difficult to conceptualise how systemic 3D profiling can relate ICI sensitivity to the relevant aspects of the immune microenvironment in the tumour tissue, specifically the impact of tumour burden and various mutations which may contribute to the individual patient outcome. Important insights into the complex relationships between systemic and localised deregulations at the level of 3D genomic architecture comes from the latest breakthroughs into the understanding of mechanisms behind systemic epigenetic synchronisation; from horizontal transfer of secretory microRNA in cancer [70], to cancer exosomes promoting tumorigenesis [71], to characterisation of cross-tissue genetic-epigenetic synchronisation effects [72]. It has been noted that almost half of the epigenetic bandwidth reflects tissue-specific patterns while the other half reflects systemic synchronised patterns, indicating molecular pathological dysregulation from the distant sites of origin, including even on the other side of the brain barrier [72].

One of the most relevant studies has demonstrated that two independent methodologies of exposure to exosome signalling from prostate cancer cells leads to changes in 3D genomic profiles of the monocytes, consistent with the clinical biomarkers already identified in prostate cancer patients [44,51,73].

Exosome traffic, or in its broader definition—the traffic of extracellular vesicles (EV), is a high-density flow (up to 10^10^ per mL of blood) of endosomal representations from multiple cells of origin in the body, loaded with selective protein content (including PD-1 and PD-L1), metabolites, and non-coding RNAs—all of which could potentially act as epigenetic regulators in the recipient effector cells (Figure 10). Here, the exosome traffic has been clearly observed to switch regulatory 3D architecture towards a cell of origin profile on a subset of genomic loci [73,74,75]. Such changes in 3D architecture of the genome are highly informative of the primary site conditions. Moreover, they have been shown to take place as similar stable binary changes both in cell–cell and cell–conditioned media settings. Together, this has provided the first insights into a mechanistic explanation of the concordance observed between individual 3D genomic switches detected in the blood of patients and within their primary tumours [73].

The current study represents a proof of concept that 3D genomic changes, measurable in blood, can be used as biomarkers of response prediction to PD-(L)1 ICIs in oncology. In this study, all patients underwent ICI treatment, with no comparator arm undergoing control treatment or basic standard of care. The single-arm design of this study does not allow us to definitively differentiate between predictive and prognostic values of the classifier. Extension of this work to a larger number of advanced patients with different ICI therapies and with a comparator treatment arm could help further validate the predictive value of the developed EpiSwitch ICI biomarker classifier.

The endpoint of the model developed is whether an individual patient, rather than just a localised neoplasm, will respond to checkpoint inhibition. This is an assessment of the whole biological system, capturing the dynamic between the systemic host responses and profile of the specific features of the tumour mass itself. The analyte we measure, a present or absent conditional chromosome conformation has certain benefit over existing continuous data modalities (RNA and protein expression levels, DNA methylation, IHC etc.), due to it being a binary marker at its root. There is similarity through its binary nature with classic genetic risk markers like SNPs. However, a SNP represents a single point of change in 3.2 billion full genomic length, with the potential for four base differences. Such features significantly reduce the effect size for SNPs as a biomarker modality. One requires a high number of input samples in order to identify any significant SNPs of interest. The conditional chromosome conformations are also binary, they establish themselves over an extensive footprint of anchor site sequences and demonstrate lasting stability, featuring a high effect size as biomarkers. Importantly, the high effect size helps to reduce the bias potential when using mid-sized heterogenic populations.

Currently, further collections and monitoring of IO responses by RECIST 1.1 and by the identified EpiSwitch predictive biomarkers have been expanded to over five hospital sites and clinical practices, from Malaysia to the US. Real life data on predictive stratifications for IO patients will further evaluate the established classification model. A pan-therapy application of the test with response prediction across tumour types with historically low ORRs could help to improve both patient outcomes and increase the cost effectiveness of cancer care [76,77].

## 5. Conclusions

With the rapid advancement of novel therapies targeting the immune checkpoint pathway, there is a pressing need to develop better biomarkers to assess likely clinical response in advance of therapeutic intervention. Here, we report on a novel 3D genomics approach to identify predictive blood-based markers that can identify, with high accuracy, individuals that are likely to respond to PD-(L)1 ICIs monotherapy, especially across tumour types with low ORRs. The 3D genomics approach described here has been developed into a clinical assay to assist in treatment decisions, help improve patient selection for optimised treatment, help better utilise alternative effective treatments, minimise or avoid unnecessarily toxicity, and efficiently manage costs and resources.

## Figures and Tables

**Figure 1 cancers-15-02696-f001:**
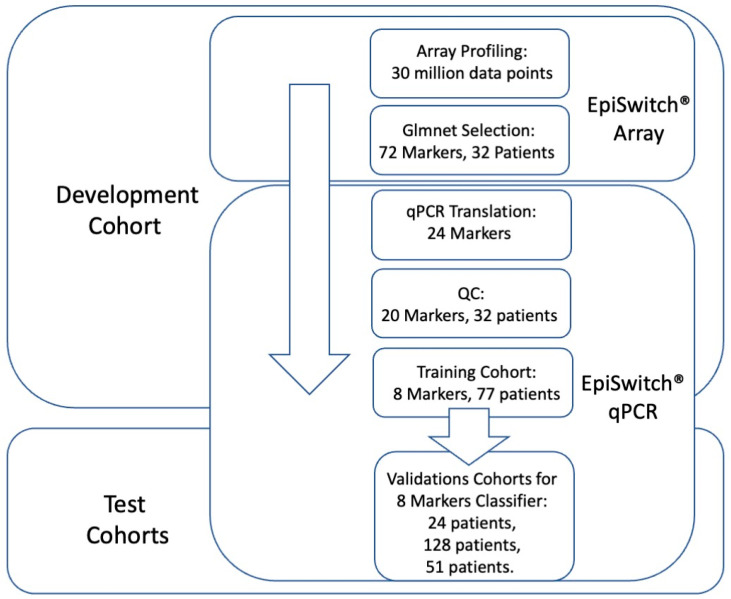
Workflow for development and testing the 3D genomic classifier model for prediction of ICI treatment response outcome. From EpiSwitch array screening profiles accounting for over 30 million data points, 72 top array markers were selected based on Glmnet logistic regression. Twenty-four markers qualified for translation into EpiSwitch qPCR format, of which 20 markers have passed quality control and feature reduction control on 32 patient samples. A training cohort of 77 patients was used to build a predictive classifier model based on eight qPCR markers. It was then validated on independent cohorts of 24, 128, and 51 patients.

**Figure 2 cancers-15-02696-f002:**
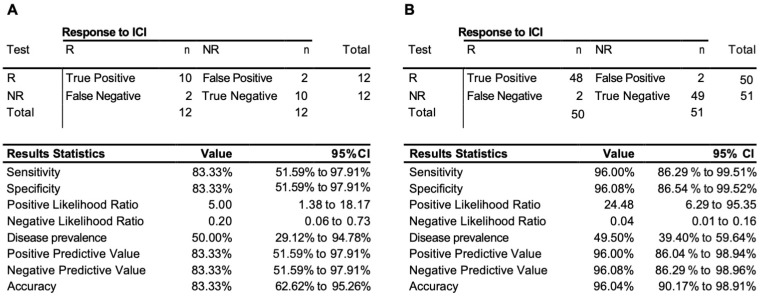
Performance of the prognostic biomarker classifier for calling response (R) and non-response (NR) to ICI treatment outcome. (**A**) Confusion matrix and test performance statistics for the eight-marker classifier on the 24 patients of the test cohort and (**B**) in the combined train and test cohort of 102 patients.

**Figure 3 cancers-15-02696-f003:**
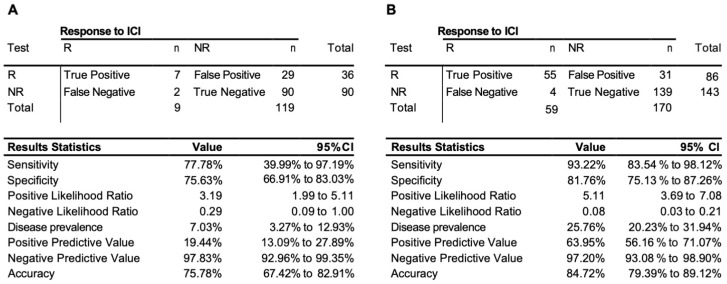
Performance of the prognostic biomarker classifier for calling response (R) and non-response (NR) to ICI treatment outcome. (**A**) Confusion matrix and test performance statistics for the eight-marker classifier on the 128 patients of the test cohort 2 and (**B**) in the combined train and test cohort of 229 patients.

**Figure 4 cancers-15-02696-f004:**
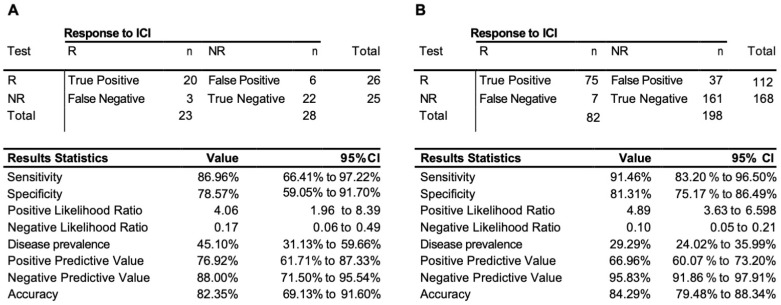
Performance of the prognostic biomarker classifier for calling response (R) and non-response (NR) to ICI treatment outcome. (**A**) Confusion matrix and test performance statistics for the eight-marker classifier on the 51 patients of the observational test cohort #3 and (**B**) in the total combined train and test cohort of 280 patients.

**Figure 5 cancers-15-02696-f005:**
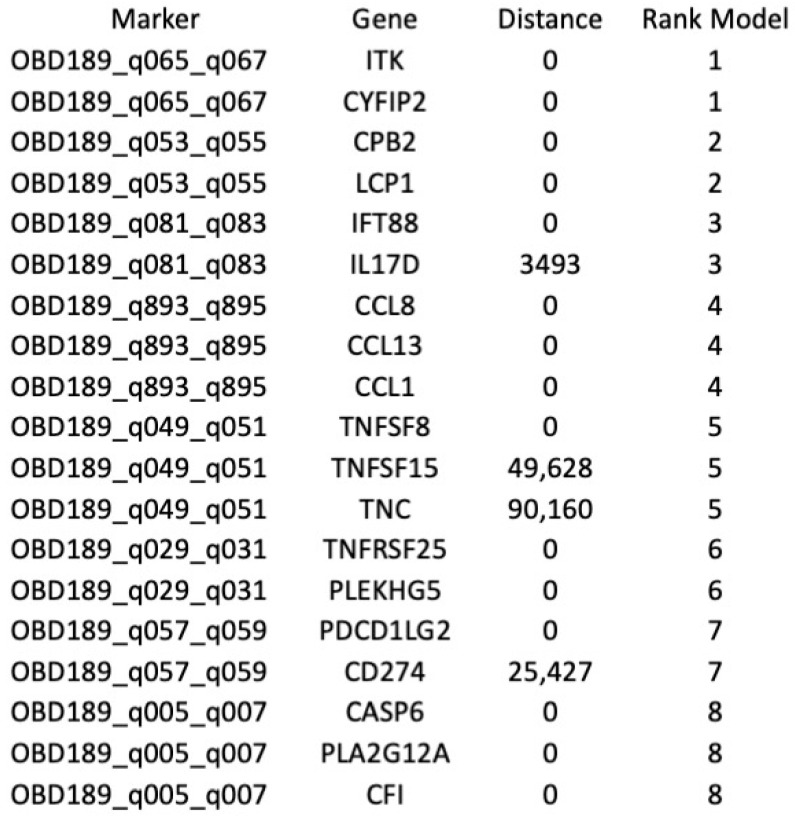
Mapping of the top eight predictive 3D genetic markers. Identities of genes associated with the location of the eight top 3D genomic markers analysed in this study. Distance—actual distance of the 3D long-range interaction marker from the gene ORF. Rank_Model—ranking order of the markers in the classifier model.

**Figure 6 cancers-15-02696-f006:**
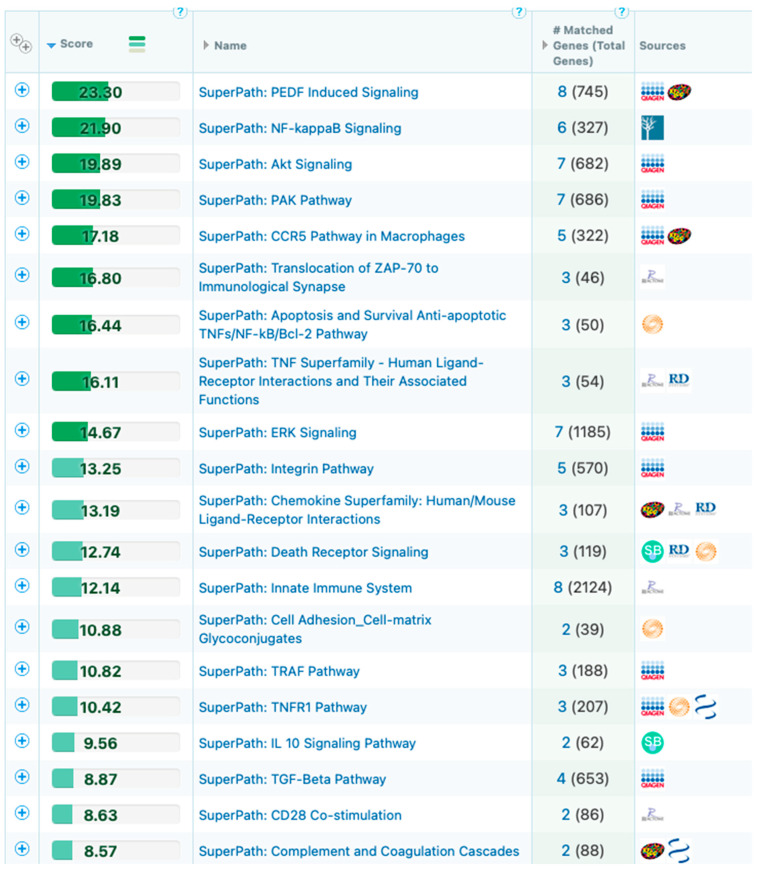
Mapping of the top eight predictive 3D genomic markers to biological pathways. Analysis of the top eight 3D genomic markers separating PD-1/PD-L1 ICI responders and non-responders at baseline.

**Figure 7 cancers-15-02696-f007:**
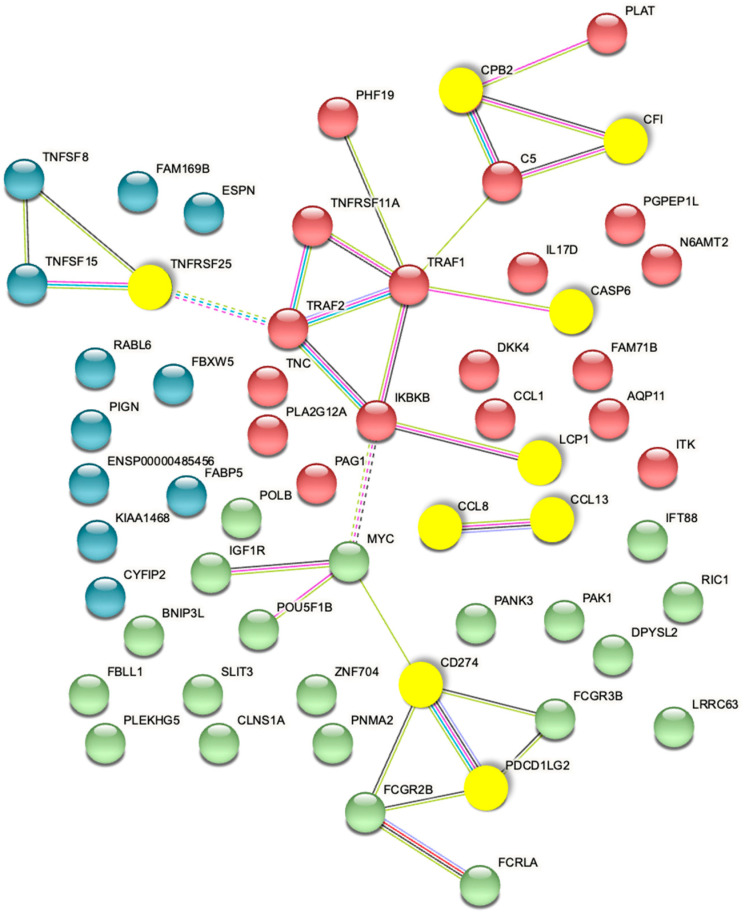
STRING Network associated with baseline prediction of response/non-response to ICI treatment. The proteins encoded by genes in the top 3D genomic markers associated with baseline response/non-response to ICI profile show a highly connected network. The yellow nodes mark the coding regions associated with the top eight markers of the validated classifier.

**Figure 8 cancers-15-02696-f008:**
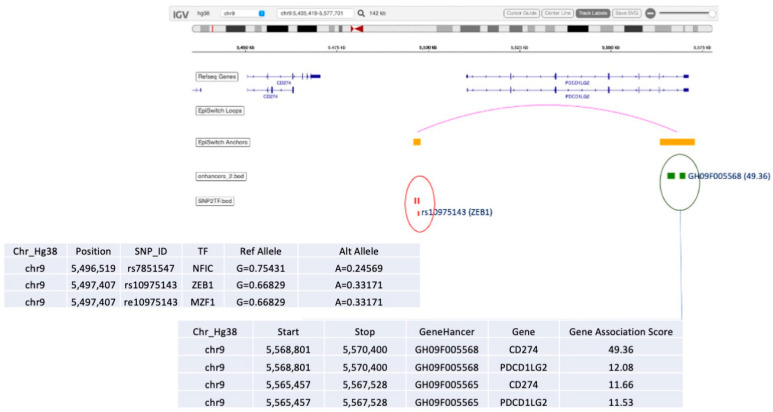
EpiSwitch Data Portal view of the EpiSwitch Marker located in CD274/PDCD1LG2 region imposed over the Integrative Genomic Viewer (IGV) (ref). Anchor sites brought into juxtaposition by chromosome long-range interactions of the EpiSwitch marker overlap with the listed SNPs and enhancers. Analysis of SNPs identified affected binding sites for transcription factors (TFs) such as ZEB1 [63]. GeneCards analysis of the enhancers identifies CD274 enhancer at the distal anchor site of the EpiSwitch marker. CD274 corresponds to PD-L1, and PDCD1LG2 to PD-L2 immune checkpoints.

**Figure 9 cancers-15-02696-f009:**
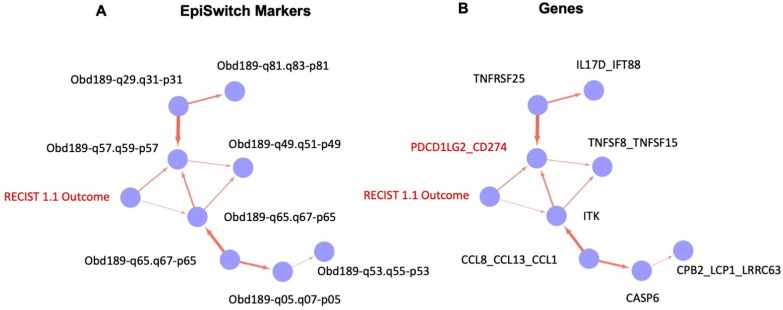
Causal graphs (Bayesian causal networks) based on training set of the EpiSwitch biomarkers and clinical outcome of IO treatment. Probabilistic relationship of the EpiSwitch markers and RECIST 1.1 assessment of IO treatment outcome, based on 3D genomic architecture (**A**) and gene identities at the sites of EpiSwitch markers (**B**). The edge in (**B**) marks the causal relationship between RECIST 1.1 outcome and PD-L1 (CD274)-based EpiSwitch marker (marked in red, see also Figure 8 for the detailed location of the marker).

**Figure 10 cancers-15-02696-f010:**
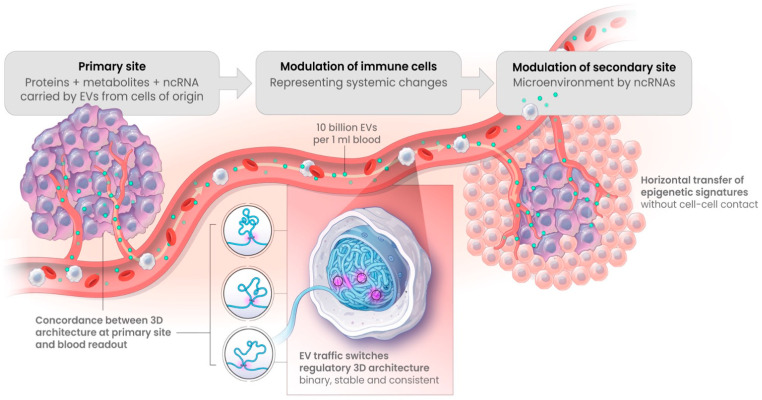
Horizontal transfer by EV, including exosomes, carry epigenetic information from primary sites to the secondary sites and the sites of systemic detection. Extracellular vehicles, including exosomes, carry representative loads of proteins, metabolites, and non-coding RNAs from primary sites, all of which could act as epigenetic regulatory factors for resetting 3D architecture of the selective genetic loci of the effector cells [73]. In early symptomatic systemic readouts, the cells of immunosurveillance were identified as a potential subject of primary 3D modulation, as in the case of early melanoma detection [39].

**Table 1 cancers-15-02696-t001:** Array based 3D genomic marker leads for baseline response to ICI.

Probe ^a^	Glmnet_Coef ^b^	*p*. Value	adj.*p*.Val ^c^	FC ^d^	Primers Design ID ^e^	Primer1 ^f^	Primer2 ^f^	Gene ^g^
Hg38_9_5495992_5572986_RR	0.247101522	0.001213344	0.039106564	1.18337794	OBD189-q0s57/q059	GAGGGTCACTCACTGCCCAACAGGC	GACTGTAAGGTAGAAATCCTGCCTGGGT	PDCD1LG2, CD274
Hg38_9_5495992_5572986_RR	0.247101522	0.001213344	0.039106564	1.18337794	OBD189-q057/q059	GAGGGTCACTCACTGCCCAACAGGC	GACTGTAAGGTAGAAATCCTGCCTGGGT	CD274
Hg38_9_5495992_5572986_RR	0.247101522	0.001213344	0.039106564	1.18337794	OBD189-q057/q059	GAGGGTCACTCACTGCCCAACAGGC	GACTGTAAGGTAGAAATCCTGCCTGGGT	RIC1
Hg38_13_20664875_20698635_FF	0.111815729	0.039447494	0.258498898	1.165419467	OBD189-q081/q083	GAAGTGCCACGAGAAGGAGGATGGTCC	GGGCTGTGTCCTGATAAACCCATTGTTA	IFT88
Hg38_13_20664875_20698635_FF	0.111815729	0.039447494	0.258498898	1.165419467	OBD189-q081/q083	GAAGTGCCACGAGAAGGAGGATGGTCC	GGGCTGTGTCCTGATAAACCCATTGTTA	IL17D
Hg38_13_20664875_20698635_FF	0.111815729	0.039447494	0.258498898	1.165419467	OBD189-q081/q083	GAAGTGCCACGAGAAGGAGGATGGTCC	GGGCTGTGTCCTGATAAACCCATTGTTA	N6AMT2
Hg38_13_46087370_46193039_RF	0.132758731	0.097937296	0.391249484	1.09303819	OBD189-q053/q055	TAGAAGCAGGGAGTAGTTGAGCAATGGG	TCTTCACTTGTGCTATTGGCTTTCCAGC	CPB2
Hg38_13_46087370_46193039_RF	0.132758731	0.097937296	0.391249484	1.09303819	OBD189-q053/q055	TAGAAGCAGGGAGTAGTTGAGCAATGGG	TCTTCACTTGTGCTATTGGCTTTCCAGC	LCP1
Hg38_13_46087370_46193039_RF	0.132758731	0.097937296	0.391249484	1.09303819	OBD189-q053/q055	TAGAAGCAGGGAGTAGTTGAGCAATGGG	TCTTCACTTGTGCTATTGGCTTTCCAGC	LRRC63
Hg38_15_98731539_98790114_FF	0.165726908	0.108961414	0.410037537	1.08563184	OBD189-q001/q003	GGCTGGTGGGAGTATTTTCAAAGAGAAC	GCTCTGTTCAAGTGGCTCTGTTCCA	IGF1R
Hg38_15_98731539_98790114_FF	0.165726908	0.108961414	0.410037537	1.08563184	OBD189-q001/q003	GGCTGGTGGGAGTATTTTCAAAGAGAAC	GCTCTGTTCAAGTGGCTCTGTTCCA	PGPEP1L
Hg38_15_98731539_98790114_FF	0.165726908	0.108961414	0.410037537	1.08563184	OBD189-q001/q003	GGCTGGTGGGAGTATTTTCAAAGAGAAC	GCTCTGTTCAAGTGGCTCTGTTCCA	FAM169B
Hg38_8_42264241_42332799_FR	0.10211307	0.160069169	0.484452574	1.079808853	OBD148_261/263	CGGTGAGCACGGTCTGTCTACTT	GTCCTGGGTCCTGGGTGAAAGTC	IKBKB
Hg38_8_42264241_42332799_FR	0.10211307	0.160069169	0.484452574	1.079808853	OBD148_261/263	CGGTGAGCACGGTCTGTCTACTT	GTCCTGGGTCCTGGGTGAAAGTC	POLB
Hg38_8_42264241_42332799_FR	0.10211307	0.160069169	0.484452574	1.079808853	OBD148_261/263	CGGTGAGCACGGTCTGTCTACTT	GTCCTGGGTCCTGGGTGAAAGTC	DKK4
Hg38_1_6461604_6515315_FR	0.049660907	0.171178937	0.498342453	1.125005092	OBD189-q029/q031	TGCCCGTCGTGGTTCCGCCTTCA	AGAGACCCACCCCAGCCTCCTGA	TNFRSF25
Hg38_1_6461604_6515315_FR	0.049660907	0.171178937	0.498342453	1.125005092	OBD189-q029/q031	TGCCCGTCGTGGTTCCGCCTTCA	AGAGACCCACCCCAGCCTCCTGA	PLEKHG5
Hg38_1_6461604_6515315_FR	0.049660907	0.171178937	0.498342453	1.125005092	OBD189-q029/q031	TGCCCGTCGTGGTTCCGCCTTCA	AGAGACCCACCCCAGCCTCCTGA	ESPN
Hg38_4_109703339_109741090_RF	0.050864625	0.218283899	0.551427137	1.109210269	OBD189-q005/q007	CCCCAACTCACAACACCCCAGAC	AGAGGAGGGCAAGGTGTCTGGCT	CASP6
Hg38_4_109703339_109741090_RF	0.050864625	0.218283899	0.551427137	1.109210269	OBD189-q005/q007	CCCCAACTCACAACACCCCAGAC	AGAGGAGGGCAAGGTGTCTGGCT	PLA2G12A
Hg38_4_109703339_109741090_RF	0.050864625	0.218283899	0.551427137	1.109210269	OBD189-q005/q007	CCCCAACTCACAACACCCCAGAC	AGAGGAGGGCAAGGTGTCTGGCT	CFI
Hg38_8_81007411_81099880_FR	0.115763362	0.224881246	0.55802192	1.062648413	OBD189-q061/q063	TGGACAGCCACTACTCAACCTTTTCCTA	CAAACCCAGATTGGACCTCACAGCCCC	PAG1
Hg38_8_81007411_81099880_FR	0.115763362	0.224881246	0.55802192	1.062648413	OBD189-q061/q063	TGGACAGCCACTACTCAACCTTTTCCTA	CAAACCCAGATTGGACCTCACAGCCCC	ZNF704
Hg38_8_81007411_81099880_FR	0.115763362	0.224881246	0.55802192	1.062648413	OBD189-q061/q063	TGGACAGCCACTACTCAACCTTTTCCTA	CAAACCCAGATTGGACCTCACAGCCCC	FABP5
Hg38_1_161633494_161661864_RF	0.027579594	0.247573207	0.581047746	1.148972903	OBD189-q041/q043	TTGCCACCTGTCTCAGATACCCTTGGTT	GCTGCTCCTCTTGCCTGGAATGCCTATT	FCGR2B
Hg38_1_161633494_161661864_RF	0.027579594	0.247573207	0.581047746	1.148972903	OBD189-q041/q043	TTGCCACCTGTCTCAGATACCCTTGGTT	GCTGCTCCTCTTGCCTGGAATGCCTATT	FCGR3B
Hg38_1_161633494_161661864_RF	0.027579594	0.247573207	0.581047746	1.148972903	OBD189-q041/q043	TTGCCACCTGTCTCAGATACCCTTGGTT	GCTGCTCCTCTTGCCTGGAATGCCTATT	FCRLA
Hg38_5_157178319_157271762_RR	0.026456597	0.298498007	0.628568025	1.13173441	OBD189-q065/q067	TGTATGTCTCCTGAGGTGAAGCAAGAGG	CTTCCACCGTGCCCGCAGCCAGC	ITK
Hg38_5_157178319_157271762_RR	0.026456597	0.298498007	0.628568025	1.13173441	OBD189-q065/q067	TGTATGTCTCCTGAGGTGAAGCAAGAGG	CTTCCACCGTGCCCGCAGCCAGC	CYFIP2
Hg38_5_157178319_157271762_RR	0.026456597	0.298498007	0.628568025	1.13173441	OBD189-q065/q067	TGTATGTCTCCTGAGGTGAAGCAAGAGG	CTTCCACCGTGCCCGCAGCCAGC	FAM71B
Hg38_9_114957908_114977746_RF	−0.026269829	0.301172707	0.630926161	−1.124920787	OBD148-q917/q919	TTGCTTGTGAGTTTGATGCAG	AAGCCAAATGGGCCTAGCCA	TNFSF8
Hg38_9_114957908_114977746_RF	−0.026269829	0.301172707	0.630926161	−1.124920787	OBD148-q917/q919	TTGCTTGTGAGTTTGATGCAG	AAGCCAAATGGGCCTAGCCA	TNC
Hg38_9_114957908_114977746_RF	−0.026269829	0.301172707	0.630926161	−1.124920787	OBD148-q917/q919	TTGCTTGTGAGTTTGATGCAG	AAGCCAAATGGGCCTAGCCA	TNFSF15
Hg38_18_62330039_62362521_FR	−0.102570352	0.301374789	0.631092221	−1.070331563	OBD189-q037/q039	CCTACTGGCACCACTGTGTTGGCTGG	TATCATAATCAGGCAACTGGCTGGTGC	TNFRSF11A
Hg38_18_62330039_62362521_FR	−0.102570352	0.301374789	0.631092221	−1.070331563	OBD189-q037/q039	CCTACTGGCACCACTGTGTTGGCTGG	TATCATAATCAGGCAACTGGCTGGTGC	KIAA1468
Hg38_18_62330039_62362521_FR	−0.102570352	0.301374789	0.631092221	−1.070331563	OBD189-q037/q039	CCTACTGGCACCACTGTGTTGGCTGG	TATCATAATCAGGCAACTGGCTGGTGC	PIGN
Hg38_9_136904007_136941363_RF	0.021876797	0.302188532	0.631834649	1.115259064	OBD189-q009/q011	AGCACTCGTCGTTGGGCGTGTAG	CGGCACACCTCTACTCTCAGCCT	RABL6
Hg38_9_136904007_136941363_RF	0.021876797	0.302188532	0.631834649	1.115259064	OBD189-q009/q011	AGCACTCGTCGTTGGGCGTGTAG	CGGCACACCTCTACTCTCAGCCT	TRAF2
Hg38_9_136904007_136941363_RF	0.021876797	0.302188532	0.631834649	1.115259064	OBD189-q009/q011	AGCACTCGTCGTTGGGCGTGTAG	CGGCACACCTCTACTCTCAGCCT	FBXW5
Hg38_17_34316073_34373948_RF	−0.243856053	0.359171973	0.67918219	−1.036704719	OBD148-q893/q895	ACTTGTGGCTTCCTTAGCCC	TCCTTTGCAGGTATGGACATC	CCL8
Hg38_17_34316073_34373948_RF	−0.243856053	0.359171973	0.67918219	−1.036704719	OBD148-q893/q895	ACTTGTGGCTTCCTTAGCCC	TCCTTTGCAGGTATGGACATC	CCL13
Hg38_17_34316073_34373948_RF	−0.243856053	0.359171973	0.67918219	−1.036704719	OBD148-q893/q895	ACTTGTGGCTTCCTTAGCCC	TCCTTTGCAGGTATGGACATC	CCL1
Hg38_13_20664875_20744490_FR	0.080919132	0.460746676	0.752684253	1.036210668	OBD189-q073/q075	GGAAGTGCCACGAGAAGGAGGATGGTCC	GGTAAGATGAGGCTGTGGGCAAGGAGC	IFT88
Hg38_13_20664875_20744490_FR	0.080919132	0.460746676	0.752684253	1.036210668	OBD189-q073/q075	GGAAGTGCCACGAGAAGGAGGATGGTCC	GGTAAGATGAGGCTGTGGGCAAGGAGC	IL17D
Hg38_13_20664875_20744490_FR	0.080919132	0.460746676	0.752684253	1.036210668	OBD189-q073/q075	GGAAGTGCCACGAGAAGGAGGATGGTCC	GGTAAGATGAGGCTGTGGGCAAGGAGC	N6AMT2
Hg38_11_77430379_77519103_RF	0.150008419	0.482939086	0.766977325	1.027489411	OBD189-q033/q035	CATAACCACACTGCTACCAACACACCTA	CTGGTTATTCGGACACTCATAGGACTGG	PAK1
Hg38_11_77430379_77519103_RF	0.150008419	0.482939086	0.766977325	1.027489411	OBD189-q033/q035	CATAACCACACTGCTACCAACACACCTA	CTGGTTATTCGGACACTCATAGGACTGG	CLNS1A
Hg38_11_77430379_77519103_RF	0.150008419	0.482939086	0.766977325	1.027489411	OBD189-q033/q035	CATAACCACACTGCTACCAACACACCTA	CTGGTTATTCGGACACTCATAGGACTGG	AQP11
Hg38_8_42264241_42292124_FR	0.028350069	0.486591863	0.769305711	1.0425716	OBD189-q025/q027	GGTGAGCACGGTCTGTCTACTTTCCC	GGACCCAGGCTCTGCTGCTACAG	IKBKB
Hg38_8_42264241_42292124_FR	0.028350069	0.486591863	0.769305711	1.0425716	OBD189-q025/q027	GGTGAGCACGGTCTGTCTACTTTCCC	GGACCCAGGCTCTGCTGCTACAG	POLB
Hg38_8_42264241_42292124_FR	0.028350069	0.486591863	0.769305711	1.0425716	OBD189-q025/q027	GGTGAGCACGGTCTGTCTACTTTCCC	GGACCCAGGCTCTGCTGCTACAG	PLAT
Hg38_18_62296384_62386748_FF	0.055316179	0.499574836	0.777354809	1.031115956	OBD189-q045/q047	CATAGACCCAGGTGTGCTCCGTGGCAGC	GAGCACTGGTTCCCCGCAAATACTGGG	KIAA1468
Hg38_18_62296384_62386748_FF	0.055316179	0.499574836	0.777354809	1.031115956	OBD189-q045/q047	CATAGACCCAGGTGTGCTCCGTGGCAGC	GAGCACTGGTTCCCCGCAAATACTGGG	TNFRSF11A
Hg38_18_62296384_62386748_FF	0.055316179	0.499574836	0.777354809	1.031115956	OBD189-q045/q047	CATAGACCCAGGTGTGCTCCGTGGCAGC	GAGCACTGGTTCCCCGCAAATACTGGG	PIGN
Hg38_9_114855753_114929419_FR	0.134189513	0.52698868	0.793876125	1.033468601	OBD189-q049/q051	CCATTGTTGCTCAGGCTGCCCTCTTGC	GCATTCAAGTGACAGAGAGAAAAGAGGC	TNFSF8
Hg38_9_114855753_114929419_FR	0.134189513	0.52698868	0.793876125	1.033468601	OBD189-q049/q051	CCATTGTTGCTCAGGCTGCCCTCTTGC	GCATTCAAGTGACAGAGAGAAAAGAGGC	TNFSF15
Hg38_9_114855753_114929419_FR	0.134189513	0.52698868	0.793876125	1.033468601	OBD189-q049/q051	CCATTGTTGCTCAGGCTGCCCTCTTGC	GCATTCAAGTGACAGAGAGAAAAGAGGC	TNC
Hg38_8_26561792_26644530_FR	0.055843287	0.625162161	0.848039488	1.042292141	OBD189-q069/q071	CAGTATGAGTGTTCTGTGGCTGCTCCCA	GCGTGTCTCTCAGGGAAGGCAGGATGC	DPYSL2
Hg38_8_26561792_26644530_FR	0.055843287	0.625162161	0.848039488	1.042292141	OBD189-q069/q071	CAGTATGAGTGTTCTGTGGCTGCTCCCA	GCGTGTCTCTCAGGGAAGGCAGGATGC	PNMA2
Hg38_8_26561792_26644530_FR	0.055843287	0.625162161	0.848039488	1.042292141	OBD189-q069/q071	CAGTATGAGTGTTCTGTGGCTGCTCCCA	GCGTGTCTCTCAGGGAAGGCAGGATGC	BNIP3L
Hg38_8_127691489_127740424_FR	−0.000580983	0.676617738	0.873913467	1.034088403	OBD189-q013/q015	GTCACCTTCATCTCCTTCTCACAGCAG	GCTTCGCTTACCAGAGTCGCTGC	MYC
Hg38_8_127691489_127740424_FR	−0.000580983	0.676617738	0.873913467	1.034088403	OBD189-q013/q015	GTCACCTTCATCTCCTTCTCACAGCAG	GCTTCGCTTACCAGAGTCGCTGC	AC108925.1
Hg38_8_127691489_127740424_FR	−0.000580983	0.676617738	0.873913467	1.034088403	OBD189-q013/q015	GTCACCTTCATCTCCTTCTCACAGCAG	GCTTCGCTTACCAGAGTCGCTGC	POU5F1B
Hg38_9_120888366_120919710_RR	0.018753385	0.784545989	0.922028968	1.010214644	OBD189-q017/q019	CCCAGTTGTCCAGGTTGCTGCCT	CCTGGAGCAGAACCTGTCAGACC	PHF19
Hg38_9_120888366_120919710_RR	0.018753385	0.784545989	0.922028968	1.010214644	OBD189-q017/q019	CCCAGTTGTCCAGGTTGCTGCCT	CCTGGAGCAGAACCTGTCAGACC	TRAF1
Hg38_9_120888366_120919710_RR	0.018753385	0.784545989	0.922028968	1.010214644	OBD189-q017/q019	CCCAGTTGTCCAGGTTGCTGCCT	CCTGGAGCAGAACCTGTCAGACC	C5
Hg38_13_20664875_20691044_FF	−0.023912747	0.80365095	0.929901824	−1.012741717	OBD189-q077/q079	GGAAGTGCCACGAGAAGGAGGATGGTCC	CCACCCAGTTCCTCCAGGCATAGCAGG	IFT88
Hg38_13_20664875_20691044_FF	−0.023912747	0.80365095	0.929901824	−1.012741717	OBD189-q077/q079	GGAAGTGCCACGAGAAGGAGGATGGTCC	CCACCCAGTTCCTCCAGGCATAGCAGG	IL17D
Hg38_13_20664875_20691044_FF	−0.023912747	0.80365095	0.929901824	−1.012741717	OBD189-q077/q079	GGAAGTGCCACGAGAAGGAGGATGGTCC	CCACCCAGTTCCTCCAGGCATAGCAGG	N6AMT2
Hg38_5_168579937_168620163_RR	0.019685283	0.8953453	0.964751362	−1.005369684	OBD189-q021/q023	CCGACCCTAACATTCAAGGTGTCTCTAT	GAGTCAGCGTGTAGTGCTCCCAC	PANK3
Hg38_5_168579937_168620163_RR	0.019685283	0.8953453	0.964751362	−1.005369684	OBD189-q021/q023	CCGACCCTAACATTCAAGGTGTCTCTAT	GAGTCAGCGTGTAGTGCTCCCAC	SLIT3
Hg38_5_168579937_168620163_RR	0.019685283	0.8953453	0.964751362	−1.005369684	OBD189-q021/q023	CCGACCCTAACATTCAAGGTGTCTCTAT	GAGTCAGCGTGTAGTGCTCCCAC	FBLL1

^a^ Internal Array probe ID. ^b^ Glment coefficient. ^c^ Adjusted *p* value. ^d^ Fold Change. ^e^ Internal primer design ID. ^f^ Primer sequence. ^g^ Gene at the locus of the probe. Top array-based 3D genomic markers identified from over 30 million data points in 32 responders and non-responders across several indications and several choices of ICI: Pembrolizumab, Atezolizumab, or Durvalumab. Probe—array-based marker coordinates for long-range interaction junction; Glmnet coefficient, *p* value, Adjusted *p* value, and Fold Change (FC) are array-based measures of markers in the comparison of responder and non-responder groups; primer design ID—qPCR primer probe designs corresponding to array probes. Twenty-four optimal in silico designs (marked in green) were taken forward to quality control checks in temperature gradient analysis. Gene—identity of genes of interest in the location of 3D genomic markers.

**Table 2 cancers-15-02696-t002:** Translation and selection of top markers into qPCR format.

Episwitch Interaction ^a^	Primers Design ID ^b^	Probe Used	Opt Ann Tm ^c^	QC Optimisation ^d^	Screen 1	Screen 2
CASP6_4_109703339_109741090_RF	OBD189-q005/q007	OBD189-p005	68	Passed	Passed	Passed
IGF1R_15_98731539_98790114_FF	OBD189-q001/q003	OBD189-p003	64.4	Passed		Failed
IKBKB_8_42264241_42292124_FR	OBD189-q025/q027	OBD189-p025	66.4	Passed		Failed
IKBKB_8_42264241_42332799_FR	OBD148_261/263	OBD189-p261	68	Passed		Failed
IL17D_13_20664875_20691044_FF	OBD189-q077/q079	OBD189-p077	67.5	Passed		Failed
ITK_5_157178319_157271762_RR	OBD189-q065/q067	OBD189-p065	66.4	Passed	Passed	Passed
MYC_8_127691489_127740424_FR	OBD189-q013/q015	OBD189-p013	66.4	Passed		Failed
ORF102_17_34316073_34373948_RF	OBD148-q893/q895	OBD148-p893	62	Passed	Passed	Passed
ORF197_8_26561792_26644530_FR	OBD189-q069/q071	N/A ^e^	N/A	Failed		Failed
ORF243_1_161633494_161661864_RF	OBD189-q041/q043	OBD189-p043	62	Passed		Failed
ORF313_13_20664875_20698635_FF	OBD189-q081/q083	OBD189-p081	64.4	Passed	Passed	Passed
ORF313_13_20664875_20744490_FR	OBD189-q073/q075	N/A	N/A	Failed		Failed
ORF369_13_46087370_46193039_RF	OBD189-q053/q055	OBD189-p053	67.5	Passed	Passed	Passed
ORF479_8_81007411_81099880_FR	OBD189-q061/q063	OBD189-p061	64.4	Passed		Failed
ORF480_11_77430379_77519103_RF	OBD189-q033/q035	OBD189-p033	66.4	Passed		Failed
ORF482_5_168579937_168620163_RR	OBD189-q021/q023	OBD189-p021	66.4	Passed		Failed
ORF698_18_62296384_62386748_FF	OBD189-q045/q047	OBD189-p045	66.4	Passed		Failed
ORF698_18_62330039_62362521_FR	OBD189-q037/q039	N/A	N/A	Failed		Failed
ORF703_1_6461604_6515315_FR	OBD189-q029/q031	OBD189-p031	67.5	Passed	Passed	Passed
ORF705_9_114855753_114929419_FR	OBD189-q049/q051	OBD189-p049	66.4	Passed	Passed	Passed
ORF712_9_120888366_120919710_RR	OBD189-q017/q019	OBD189-p017	68	Passed		Failed
PDCD1LG2_9_5495992_5572986_RR	OBD189-q057/q059	OBD189-p057	67.5	Passed	Passed	Passed
TNFSF8_9_114957908_114977746_RF	OBD148-q917/q919	N/A	N/A	Failed		Failed
TRAF2_9_136904007_136941363_RF	OBD189-q009/q011	OBD189-p009	64.4	Passed		Failed

^a^ Internal EpiSwitch Interaction ID. ^b^ Internal primer design ID. ^c^ Optimum Annealing Temperature. ^d^ Quality Control Optimisation. ^e^ Not Available. Temperature gradient optimisation for primer probe designs of the top 24 markers: markers that failed quality control and feature reduction in Screen 1 and Screen 2, are marked.

## Data Availability

The datasets used and analysed during the current study are available from the corresponding author on reasonable request.

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
