# Peer review of "Development and Validation of Blood-Based Predictive Biomarkers for Response to PD-1/PD-L1 Checkpoint Inhibitors: Evidence of a Universal Systemic Core of 3D Immunogenetic Profiling across Multiple Oncological Indications"

_cancers, 2023, doi:10.3390/cancers15102696_

Round 1
Reviewer 1 Report
No particular correction/implementations needed.
The paper is good for publication
Reviewer 2 Report
The manuscript from E Hunter et al describes a very preliminary assessment on the potential of a 3D genomic approach to identify patients receiving major benefit from Immune checkpont treatment of human cancer
The experimental design is sound but the study interpretation is limited by the quite low population of enrolled patients as well as their heterogeneity making the risk of bias quite high.
Moreover it is is hard to coceptualize how PBMNC nuclei 3D genomics can capture factors driving ICI inhibitors sensitivity which are related to the immune microenvironment in the tumor tissue and lymphoid organs as well as in tumor burden and quality of mutation which may activate the immune system. Thiese points need wider discussion, as well as some additional details on follow up studies to reach a validated biomarker status need to be provided in the revised version.
In any case an interesting paper to quickly improve in clarifying the proof of concept clinical relevance
The in silico work is of true interest
Reviewer 3 Report
Well-conducted prospective observational study of biospecimens from n=229 undergoing treatment with immune checkpoint inhibitors for cancer, to develop a biomarker model for response to immune checkpoint therapy which achieved a high accuracy of 85%, sensitivity of 93% and specificity of 82%. Novel and important results with high likelihood of translation to clinical care.
Round 2
Reviewer 2 Report
The paper has been correctly revised